# Improved Electrochemical Hydrogen Peroxide Detection Using a Nickel(II) Phthalimide-Substituted Porphyrazine Combined with Various Carbon Nanomaterials

**DOI:** 10.3390/nano13050862

**Published:** 2023-02-25

**Authors:** Amanda Leda, Mina Hassani, Tomasz Rebis, Michal Falkowski, Jaroslaw Piskorz, Dariusz T. Mlynarczyk, Peter McNeice, Grzegorz Milczarek

**Affiliations:** 1Institute of Chemistry and Technical Electrochemistry, Faculty of Chemical Technology, Poznan University of Technology, Berdychowo 4, 60-965 Poznan, Poland; 2Department of Medicinal Chemistry, Collegium Medicum in Bydgoszcz, Faculty of Pharmacy, Nicolaus Copernicus University in Torun, Dr. A. Jurasza 2, 85-089 Bydgoszcz, Poland; 3Chair and Department of Inorganic and Analytical Chemistry, Poznan University of Medical Sciences, Rokietnicka 3, 60-806 Poznan, Poland; 4Chair and Department of Chemical Technology of Drugs, Poznan University of Medical Sciences, Grunwaldzka 6, 60-780 Poznan, Poland; 5Faculty of Science and Engineering, Synthetic Organic Chemistry—Stratingh Institute of Chemistry and Chemical Building Blocks Consortium (CBBC), Nijenborgh 4, 9747 AG Groningen, The Netherlands

**Keywords:** porphyrazine, nickel, carbon nanomaterials, voltammetry, H_2_O_2_ sensor, electrocatalysis

## Abstract

A metal-free porphyrazine derivative with peripheral phthalimide substituents was metallated with a nickel(II) ion. The purity of the nickel macrocycle was confirmed using HPLC, and characterized by MS, UV–VIS, and 1D (^1^H, ^13^C) and 2D (^1^H–^13^C HSQC, ^1^H–^13^C HMBC, ^1^H–^1^H COSY) NMR techniques. The novel porphyrazine was combined with various carbon nanomaterials, such as carbon nanotubes—single walled (SWCNTs) and multi-walled (MWCNTs), and electrochemically reduced graphene oxide (rGO), to create hybrid electroactive electrode materials. The carbon nanomaterials’ effect on the electrocatalytic properties of nickel(II) cations was compared. As a result, an extensive electrochemical characterization of the synthesized metallated porphyrazine derivative on various carbon nanostructures was carried out using cyclic voltammetry (CV), chronoamperometry (CA), and electrochemical impedance spectroscopy (EIS). An electrode modified with carbon nanomaterials GC/MWCNTs, GC/SWCNTs, or GC/rGO, respectively, was shown to have a lower overpotential than a bare glassy carbon electrode (GC), allowing for the measurement of hydrogen peroxide in neutral conditions (pH 7.4). It was shown that among the tested carbon nanomaterials, the modified electrode GC/MWCNTs/**Pz3** exhibited the best electrocatalytic properties in the direction of hydrogen peroxide oxidation/reduction. The prepared sensor was determined to enable a linear response to H_2_O_2_ in concentrations ranging between 20–1200 µM with the detection limit of 18.57 µM and sensitivity of 14.18 µA mM^−1^ cm^−2^. As a result of this research, the sensors produced here may find use in biomedical and environmental applications.

## 1. Introduction

Hydrogen peroxide is a molecule that has gained a lot of attention due to its significant role in biological signaling and as a side product of some of the oxidative processes in cell metabolism [1]. It is also an important factor in industry and manufacturing, including textiles, foodstuffs, and mining [2]. Hydrogen peroxide can be detected and monitored by a series of methods: spectral analysis, colorimetry, fluorescence and luminescence analyses, and various chromatographic, titrimetric, and electrochemical methods [3,4]. Among these methods, electrochemical sensing offers certain advantages, such as a low manufacturing cost of the sensors and high sensitivity and selectivity. Unfortunately, current sensors often suffer from slow kinetics and are vulnerable to interference induced by other electroactive substances in real-life samples. Hence, the development of functional hybrid nanomaterials for the determination of hydrogen peroxide is a challenging task.

Many macrocyclic compounds have been tested for their ability to modify sensors. A comparison between different phthalocyanine complexes showed that a Ni(II)-based complex was sensitive to vapor phase electron donor sensing [5], and NiPc was confirmed to be an efficient component in devices detecting biologically important molecules [6,7]. Although the electrochemical properties of nickel(II)-containing porphyrazines (Pzs), especially sulfanyl porphyrazines, are promising, they are rarely studied [8,9]. Porphyrazines are tetrapyrrolic molecules that offer unique spectral, biological, and electrochemical features [10]. Among them, sulfanyl porphyrazines were found in recent years to exhibit interesting optical [11], photochemical [12,13], photocatalytic [14,15,16], biological [17,18,19], and electronic properties [20,21]. Their advantages include a relative ease of preparation, good solubility, and electrochemical activity [22,23]. As for the latter, porphyrinoids are well-known as potent electrocatalysts suitable for the electrochemical determination of many compounds and molecules, including H_2_O_2_, dopamine, and L-cysteine, which are known to be important for the proper functioning of the human body [24,25,26]. The aforementioned electrocatalytic features are strongly related to the metal ion in the central cavity of the macrocycle (e.g., Mn, Co, Fe, Ni). Various porphyrinoids can be covalently linked or immobilized on the surface of carbon nanostructures, including SWCNTs, MWCNTs, or graphene layers. An effective and simple strategy to boost the electrocatalytic properties of carbon nanomaterials is the non-covalent (adsorptive) attachment of a porphyrinoid macrocycle. The π-conjugative structure of carbon-based nanomaterials facilitates its interaction with porphyrinoids due to the strong π–π electronic interactions.

Taking all the above into account, the aim of this study was to fabricate an amperometric sensor for hydrogen peroxide detection. This was achieved by synthesizing a Ni(II) complex of a phthalimide-decorated sulfanyl porphyrazine, which was used to modify the electrode alongside different carbon nanomaterials, including multi-walled carbon nanotubes, single-walled carbon nanotubes, and reduced graphene oxide, among others. Based on the obtained electrochemical data, the developed sensor seems to be a promising candidate for potential biological hydrogen peroxide sensing. The proposed hybrid nanomaterials can be considered as a long-term and prospective platform for electrocatalytic hydrogen peroxide determination.

## 2. Materials and Methods

### 2.1. Synthetic Procedure for the Preparation of Metallated Porphyrazine

**2,3,7,8,12,13,17,18-Octakis[(*N*-ethylphthalimide)thio]porphyrazinato magnesium(II)** (**Pz1**) and **2,3,7,8,12,13,17,18-octakis[(*N*-ethylphthalimide)thio]porphyrazine** (**Pz2**) were synthesized following an earlier reported procedure [27].


**2,3,7,8,12,13,17,18-Octakis[(N-ethylphthalimide)thio]porphyrazinato nickel(II) (Pz3)**


Nickel(II) acetate tetrahydrate (76 mg, 0.305 mmol) and **Pz2** (120 mg, 0.061 mmol) were stirred in *N*,*N*-dimethylformamide (DMF, 15 mL) at 75 °C for 24 h. Next, after cooling to room temperature, the reaction mixture was filtered through Celite and washed with dichloromethane (150 mL). The combined filtrates were evaporated to dryness, and a dark blue residue was subjected to column chromatography: first, in the normal phase (eluents: dichloromethane/methanol, 100:1 to 20:1, *v/v*); then, on alumina (eluents: methanol, then methanol/dichloromethane 9:1 to 1:1, *v/v*) to give compound **Pz3** as a dark blue film (20 mg, 16% yield). *R_f_* (dichloromethane/methanol, 100:1, *v/v*) 0.15. UV–VIS (dichloromethane) λ_max_ nm (log ε) 303 (4.54), 352 (4.56), 667 (4.58). ^1^H NMR (400 MHz, CDCl_3_) δ, ppm: 7.27 (dd, *J* = 5.5, 3.0 Hz; C2, C5, ArH), 7.12 (dd, *J* = 5.5, 3.0 Hz; 16H, C3, C4, ArH), 4.34 (t, *J* = 6.5 Hz, 16H, SCH_2_), 4.19 (t, *J* = 6.5 Hz, 16H, NCH_2_). ^13^C NMR (100 MHz, CDCl_3_) δ, ppm: 167.8 (C=O), 148.6 (C2, C4, pyrrole C), 140.5 (C2, C3, pyrrole C), 133.7 (C3, C4, ArC), 131.5 (C1, C6, ArC), 122.9 (C2, C5, ArC), 38.3 (NCH_2_), 32.0 (SCH_2_). MS (MALDI) *m/z:* found 2011.1876, [M+H]^+^ C_96_H_65_N_16_NiO_16_S_8_ requires 2011.1878. HPLC purity 98.5–100.0% (see Appendix A).

### 2.2. Fabrication of GC/MWCNTs, GC/MWCNTs/***Pz3***, GC/SWCNTs, GC/SWCNTs/***Pz3***, GC/rGO, and GC/rGO/***Pz3*** Modified Electrodes

All materials and reagents used for the electrochemical testing are described in detail in the Appendix A, which also includes information about the apparatus and electrodes.

Prior to each electrochemical experiment, the GC electrode was polished on a polishing cloth with an aqueous suspension of Al_2_O_3_ (Buehler, 50 nm average diameter), and any impurities were subsequently removed using an ultrasonic bath containing an acetone/water solution (1:1, *v/v*). Afterwards, the cleaned surface of the GC electrode was drop-cast with 2 μL of either MWCNTs or SWCNTs dispersion (1 mg mL^−1^ in DMF), followed by oven-drying of the electrode at 60 °C until the solvent evaporated. In the case of the graphene oxide-modified electrode, initially 2 µL of the graphene oxide aqueous dispersion were drop-cast onto the electrode surface and evaporated under the same conditions. Then graphene oxide (GO) was electrochemically reduced to reduced graphene oxide (rGO) in a KH_2_PO_4_/K_2_HPO_4_ buffer (pH 7.4) in the potential ranging from 0.4 to −1.3 V (scanning rate 50 mV s^−1^). The electrochemical reduction of GO to rGO on the surface of the GC electrode is shown in Appendix A. The resulting GC/MWCNTs, GC/SWCNTs, and GC/rGO electrodes were dipped into the **Pz3** solution in dichloromethane (1 mg mL^−1^). The porphyrazines were non-covalently immobilized by soaking the respective electrodes in the **Pz3** solution. All electrodes were positioned in the desired electrolyte to conduct electrochemical testing. The glass cell holding the electrolyte was deoxygenated prior to the experiments with N_2_ gas. All electrochemical tests were performed at room temperature (around 25 °C).

## 3. Results and Discussion

### 3.1. Synthesis and Physicochemical Characterization

Phthalimide-substituted magnesium(II) Pz (**1**) and its metal-free derivative (**2**) were prepared using a previously reported three-step synthetic pathway [27]. Next, by modifying a procedure from the literature [28], a Ni^2+^ cation was introduced into the **Pz2** macrocyclic core by heating **Pz2** and nickel(II) acetate tetrahydrate in DMF, which led to the formation of nickel(II) symmetrical porphyrazine **Pz3** (Figure 1). The compounds obtained were isolated chromatographically and their properties evaluated by spectral methods—mass spectrometry and UV–VIS spectroscopy. NMR experiments were performed to confirm the structure of **Pz3**. The ^1^H and ^13^C NMR resonances were unambiguously assigned using a combination of one-dimensional (^1^H, ^13^C) and two-dimensional (^1^H–^13^C HSQC, ^1^H–^13^C HMBC and ^1^H–^1^H COSY) experiments. A detailed analysis of the NMR spectra can be found in Appendix A. The signals at 4.34 ppm were assigned to the SCH_2_ groups based on their correlation with the C2 and C3 pyrrole carbons at 140.5 ppm in the ^1^H–^13^C HMBC spectrum. The correlations between carbonyl carbon signals at 167.8 ppm and hydrogen atom signals at 4.19 ppm and 7.27 ppm allowed to assign the NCH_2_ hydrogen atoms of the ethylsulfanyl linker and phthalimide protons, respectively. The identification of aromatic proton resonances of the phthalimide moieties, as well as protons of the ethylene groups, was supported by the two-dimensional experiments (Appendix A). Furthermore, HPLC assessment of **Pz3**, performed in three different eluent systems, confirmed the purity of the new macrocycle as exceeding 98%, with detection simultaneously at 380 nm and 670 nm (see Appendix A).

The UV–VIS spectra of nickel(II) porphyrazine **Pz3** revealed the absorption maxima at 667 nm in dichloromethane and *N*,*N*-dimethylformamide, and 668 nm in dimethylsulfoxide (Figure 1). The calculated values of logarithms of the molar absorption coefficients (log ε) for these bands were 4.58 for dichloromethane and *N*,*N*-dimethylformamide and 4.57 in the case of dimethylsulfoxide (Appendix A). The comparison of the absorption spectra of **Pz3** with that of the previously obtained magnesium(II) porphyrazine **1** and demetallated analog **2** is shown in Figure 1b [27]. Notably, the Q-band absorption of nickel(II) porphyrazine **Pz3** was slightly hypsochromically shifted and much less intense than that of the magnesium(II) derivative **1** (the absorption maximum in dichloromethane reached 667 nm for **Pz3** and 674 nm for **Pz1**, while the log ε values equaled 4.58 and 4.83 for **Pz3** and **Pz1**, respectively). However, for both complexes, the Q-band was sharp and single in contrast to the broad and divided band of the demetallated derivative **2** (Figure 1b) [27].

### 3.2. Electrochemical Study of ***Pz3*** Deposited on MWCNTs, SWCNTs, and rGO

The voltammetric responses of the assigned hybrid electrodes are shown in Figure 2 for the three distinct carbon nanomaterials (MWCNTs, SWCNTs, and rGO) on which the produced porphyrazine was individually immobilized. To determine the electrochemical activity, voltammetric measurements were made in a buffered (pH 7.4) water-based electrolyte. For the bare GC electrode and MWCNTs-modified electrode, the cyclic voltamperograms acquired between −1.0 and 0.8 V vs. Ag/AgCl show conventional capacitive characteristics. The peak couples observed for GC/SWCNTs and GC/rGO at ca. 0.0 V can be ascribed to surface-confined processes involving oxygen groups attached to the surface, especially since rGO is not fully reduced and can contain various oxygen-based groups, such as quinones. After the immobilization of **Pz3**, only slight changes of CV response can be observed. However, no additional redox peaks are present. Unlike GC/SWCNTs/**Pz3** and GC/rGO/**Pz3**, the MWCNTs/**Pz3** presents significant redox features formed in the cathodic and anodic range. These redox pairs line up with the electrochemical transition of the phthalimides substituents. In our previous work, we observed the redox transition of the phtalimide groups at ca. −0.4 V in phosphate buffer electrolyte [27]. The minor peak at ca. 0.4 V should then be assigned to the oxidation of Ni^2+^ to Ni^3+^. The obtained data suggest that **Pz3** exhibits electrochemical activity only on the surface of MWCNTs. The loading of nickel can be calculated based on the charge corresponding to the voltammetric peak at ca. 0.4 V. After the integration of the anodic peak corresponding to the Ni^2+^ to Ni^3+^ transition, we estimated the metal loadings to be 11.1 ng. We can tentatively assume that the aromatic system of phthalimide offers additional π–π interactions between the porphyrazine and the MWCNT, thus attaching the **Pz3** more firmly on the surface of the modified electrode, resulting in efficient electron transfer.

Cyclic voltamperometry measurements were performed at scanning rates ranging from 10 to 100 mV s^−1^ to evaluate the kinetics of electron transport on the surface of the electrodes modified with GC/MWCNTs/**Pz3** (Figure 3). Peak currents increased linearly with scanning rate, pointing to redox activities occurring at the surface. Surface-limited redox characterization is made possible by the π–π stacking of the conjugated porphyrazine macroring with the highly delocalized π-bonding network of carbon nanomaterials.

Comparative CVs for the hybrid materials functionalized with **Pz3** (MWCNTs/**Pz3**, SWCNTs/**Pz3**, and rGO/**Pz3**) employed in this study are shown in Figure 4 in the presence of the [Fe(CN)_6_]^3−/4−^ redox couple. As seen in Figure 4A, for the GC/**Pz3** electrode, a decrease in peak current was observed in relation to the bare GC electrode. In addition, a significant increase of peak-to-peak separation from 84 to 155 mV was observed at GC/**Pz3** when compared to bare GC. This suggests hampered electron transfer kinetics at GC/**Pz3**, probably due to the insulating nature of **Pz3**. The modification of the MWCNTs resulted in an increase in the peak currents (compared to the bare GC electrode). It is well-known that MWCNTs have a highly porous structure and good electron transfer characteristics [29]. After the addition of porphyrazine to the GC/MWCNTs system, the peak-to-peak separation increased slightly. Additionally, the peak currents of the [Fe(CN)_6_]^3−/4−^ redox couple remained almost unchanged for MWCNTs/**Pz3**. Such behavior suggests that **Pz3** can strongly interact with the surface of MWCNTs via π–π stacking. As a result, electron transfer kinetics at MWCNTs/**Pz3** are fast, which is crucial for electrocatalytic sensing applications. In the case of the GC/SWCNTs/**Pz3** modified electrode, there was a decrease in both the peak current and capacitance in relation to the GC/SWCNTs electrode (Figure 4B). The same dependence can be observed for the GC/rGO and GC/rGO/**Pz3** modified electrodes (Figure 4C). Deterioration of the electrochemical properties at these two electrodes was thus observed. Table 1 summarizes the separation peak values for all tested electrodes.

The electroactive surface area was calculated by applying the Randles–Sevcik equation [30]:*I_p_* = 2.69⋯10^5^
*AD*^1/2^
*n*^3/2^
*Cv*^1/2^(1)
where, *I_p_* represents peak current [A]; *A* is the electroactive surface area of the electrode [cm^2^]; *D* represents diffusion coefficient (7.3 × 10^−6^ cm^2^ s^−1^ for [Fe(CN)_6_]^3−^); *n* is the transferred electron number (n = 1); *v* represents the scan rate [V s^−1^]; and *C* is the analyte concentration [mol cm^−3^]. Table 2 presents the results of the calculated electroactive surface areas for the modified electrodes. We can conclude that by modifying the surface of the GC electrode with the studied nanomaterials, its surface area increases. However, covering nanomaterials with **Pz3** causes a decrease in porosity, and a decrease in the electroactive surface of the electrodes is also observed.

The EIS measurements (Nyquist plots) for all the tested electrodes are shown in Figure 4D–F. The high-frequency area (semicircle in Nyquist plots) is responsible for the charge transfer resistance on the electrode surface. The relatively large semicircle observed for the GC and GC/**Pz3** electrodes (black and red lines, respectively) indicates slow electron transfer. The absence of a semicircle in Figure 4D for the GC/MWCNTs and GC/MWCNTs/**Pz3** electrodes indicates fast electron transfer, which in turn improves the electrocatalytic properties. The results are in good agreement with the cyclic voltammograms displayed in Figure 4A–C. In addition, for the electrodes GC/SWCNTs and GC/SWCNTs/**Pz3**, as well as GC/rGO and GC/rGO/**Pz3**, smaller semicircles were observed in comparison to bare GC and GC/**Pz3**.

### 3.3. The Influence of Hydrogen Peroxide on the GC, GC/***Pz3***, GC/MWCNTs, GC/MWCNTs/***Pz3***, GC/SWCNTs, GC/SWCNTs/***Pz3***, GC/rGO, and GC/rGO/***Pz3*** Electrodes

It is acknowledged that porphyrazines with transition metal ion centers are interesting candidates for the electrocatalysis of hydrogen peroxide oxidation/reduction [23]. Thus, the electrocatalytic performance of nickel(II)-containing porphyrazine (**Pz3**) was investigated in the presence of H_2_O_2_ ( Figure 5). The efficiency of the constructed hybrid platforms (GC/MWCNTs/**Pz3**, GC/SWCNTs/**Pz3**, and GC/rGO/**Pz3**) was compared with those of bare GC, GC/**Pz3,** GC/MWCNTs, GC/SWCNTs, and GC/rGO electrodes. The reduction of H_2_O_2_ may occur on unmodified GC at a very negative overpotential (Figure 5A), while the measured reductive current was rather small; an even smaller reduction current was observed in the case of the GC/**Pz3** electrode (Figure 5B). When a GC/MWCNTs electrode was employed, a slight improvement in the H_2_O_2_ redox behavior was observed (Figure 5C). In this case, both cathodic and anodic current waves were seen. Significant electrocatalytic activity was observed for the hybrid electrode of GC/MWCNTs/**Pz3**. Figure 5D (curve b) illustrates how the redox peaks were greatly enhanced when 2 mM H_2_O_2_ was added. At GC/MWCNTs/**Pz3**, hydrogen peroxide was shown to be reduced and oxidized using a small overpotential and high current. This result indicates that GC/MWCNTs/**Pz3** is a suitable electrode for hydrogen peroxide electrocatalysis. A well-defined H_2_O_2_ anode peak current was also observed, suggesting that GC/MWCNTs/**Pz3** has good electrocatalytic performance. Regarding electrode modifications, a comparison of SWCNTs and rGO (Figure 5E,G) with SWCNTs/**Pz3** and rGO/**Pz3** (Figure 5F,H) showed only a slight amplification of the peak cathode and anode currents, which indicates that SWCNTs and rGO are not promising platforms for **Pz3** immobilization in the context of H_2_O_2_ electrocatalysis.

### 3.4. Chronoamperometric Measurements of GC/MWCNTs/***Pz3***, GC/SWCNTs/***Pz3***, and GC/rGO/***Pz3*** Electrodes in the Presence of H_2_O_2_

Furthermore, under stirring conditions, chronoamperometric measurements were made for GC/MWCNTs/**Pz3**, GC/SWCNTs/**Pz3**, and GC/rGO/**Pz3**. The oxidation of H_2_O_2_ at the electrodes was driven by the applied anodic potential of +0.6 V. Figure 6A shows the increase of the amperometric signal after the addition of small amounts of H_2_O_2_. The GC/MWCNTs/**Pz3** electrode expressed linearity within 20–1200 µM of H_2_O_2_. For GC/SWCNTs/**Pz3**, linearity was seen in the H_2_O_2_ concentration range of 10 to 980 µM. For the GC/rGO/**Pz3** electrode, linearity was noted in the range of analyte concentrations from 20 to 750 µM. Figure 6B,E,H also show the amperometry results recorded for the electrode modified with only carbon-based nanomaterials—MWCNTs, SWCNTs, and rGO, respectively. In connection with the above, it was shown that the linearity in the widest range of hydrogen peroxide concentrations was recorded for the GC/MWCNTs/**Pz3** electrode. In addition, in the case of carbon nanomaterials (SWCNTs and rGO), a decrease in the current signal was observed after **Pz3** adsorption (Figure 6F,I). The estimated limits of detection (LODs) were 18.57, 9.42, and 9.15 µM for GC/MWCNTs/**Pz3**, GC/SWCNTs/**Pz3** and GC/rGO/**Pz3**, respectively, when the signal-to-noise ratio of 3 was taken into account. The limits of quantification (LOQs), which were 56.27, 28.56, and 58.04 µM, respectively, were also compared.

In Table 3, the performance of the investigated electrodes for hydrogen peroxide electroanalysis is compared with other electrodes found in the literature.

To assess the selectivity of the GC/MWCNTs/**Pz3** electrode, the amperometric response was monitored in the presence of different interferents that can commonly occur in a variety of bodily fluids (in this work: glucose, fructose, lactose, maltose, saccharose, caffeine, and sodium chloride) at +0.6 V with stirring (Figure 7). While H_2_O_2_ addition gave a fast current response, none of the inserted interferents resulted in a current reply. Therefore, the obtained sensor has a satisfactory selectivity. In addition, analogous tests were carried out for the modified electrodes: GC/SWCNTs/**Pz3** and GC/rGO/**Pz3**. The results of these analyses are shown in Appendix A. As in the case of the GC/MWCNTs/**Pz3** electrode, no interference effect was observed. A decrease in the current signal was observed over time.

## 4. Conclusions

In this study, a new phthalimide-substituted sulfanyl porphyrazine derivative possessing an Ni(II) ion in the core was obtained via chemical synthesis. The formation of this molecule was confirmed using spectral techniques, including NMR spectroscopy, mass spectrometry, and UV–VIS spectrophotometry. Additionally, detailed UV–VIS spectral studies were performed. The **Pz3** was subsequently used to etch glassy carbon electrodes, modified earlier with either multi-walled carbon nanotubes, single-walled carbon nanotubes, or reduced graphene oxide. Among the studied carbon nanomaterials, it was found that multi-walled carbon nanotubes constitute a suitable matrix for the immobilization of **Pz3** porphyrazine on the surface of the GC electrode. It has been shown that the GC/MWCNTs/**Pz3** modified electrode has excellent electrocatalytic properties in the detection of hydrogen peroxide oxidation/reduction (LOD: 18.57 μM; linear range: 20–1200 μM) and may be regarded as a potential sensor of this important molecule. The results obtained in this study show that a determination of the appropriate conductive material for electrode modification is crucial to achieving the synergistic effect and satisfactory electrocatalytic properties.

## Data Availability

Not applicable.

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
