# Peer review of "Improved Electrochemical Hydrogen Peroxide Detection Using a Nickel(II) Phthalimide-Substituted Porphyrazine Combined with Various Carbon Nanomaterials"

_nanomaterials, 2023, doi:10.3390/nano13050862_

Round 1
Reviewer 1 Report
In this article, Amanda Leda et al., synthesized a nickel containing porphyrazines denoted as Pz3, which was supported on several carbon nanomaterials including single walled carbon nanotube, multi-walled nanotube, and electrochemically reduced graphene oxide, to assess and compare their electrochemical properties as well as sensors toward H2O2. Overall, the Pz3 is new structure and well characterized, the sensor performance of GC/MWCNTs/Pz3 is impressive compared with other reported materials with a linear response range between 20-1200 μM H2O2, detection limit of 18.57 μM and sensitivity of 14.18 μA mM-1 cm-2. This paper shall be publishable in Nanomaterials. Some concern or improvement shall be addressed for further improvement:
1. I agree with author that substrate is important to determine the overall performance of sensor, however, as we all know that the substrate, particularly for carbon nanomaterials are highly determined by the synthesis process, surface functional group species, defects states, and etc. Many research groups can observe completely distinct phenomena even in the substrate like graphene or carbon nanotube. Therefore, I may trend to suggest concentrate more Pz3 molecule itself, rather compare which one is better or not. If they want to discuss more for substrate effect, they shall provide a relatively reasonable interpretation why MWCNT is best, and why the other two substrates are worse, this may inspire the readers more.
2. Line 59, the author claim that “Although the electrochemical properties of nickel (II) containing porphyrazines (Pzs), especially sulfanyl porphyrazines, are promising, there (they?) are scarcely studied”. Can the author clarify more why sulfanyl are promising or why they are less studied? Any specific role of N- ethylphthalimide in Pz3?
3. Another concern for Raman spectrum in Figure S1 is the superlattice, charge-transfer may cannot well interpret the drastic Raman shift of E2g mode in superlattice, such a phenomenon is drastically different from the reported Raman spectra in layered superlattice, (J. Am. Chem. Soc. 2017, 139, 16398−16404; Adv. Mater. 2020, 1907645; Nature 555, 231–236 (2018).) the authors shall check each Raman peak carefully and explain more.
4. Line 64 shall be their advantages are …
5. Line 163, please provide more details of Q band.
6. The caption in Figure 4 is misleading, the author should clarify the specific measurement condition for each CV spectrum. Moreover, the insets in Figure 4D, 4E, 4F are too small.
Author Response
In this article, Amanda Leda et al., synthesized a nickel containing porphyrazines denoted as Pz3, which was supported on several carbon nanomaterials including single walled carbon nanotube, multi-walled nanotube, and electrochemically reduced graphene oxide, to assess and compare their electrochemical properties as well as sensors toward H2O2. Overall, the Pz3 is new structure and well characterized, the sensor performance of GC/MWCNTs/Pz3 is impressive compared with other reported materials with a linear response range between 20-1200 μM H2O2, detection limit of 18.57 μM and sensitivity of 14.18 μA mM-1 cm-2. This paper shall be publishable in Nanomaterials. Some concern or improvement shall be addressed for further improvement:
1. I agree with author that substrate is important to determine the overall performance of sensor, however, as we all know that the substrate, particularly for carbon nanomaterials are highly determined by the synthesis process, surface functional group species, defects states, and etc. Many research groups can observe completely distinct phenomena even in the substrate like graphene or carbon nanotube. Therefore, I may trend to suggest concentrate more Pz3 molecule itself, rather compare which one is better or not. If they want to discuss more for substrate effect, they shall provide a relatively reasonable interpretation why MWCNT is best, and why the other two substrates are worse, this may inspire the readers more.
Thank you for the valuable comment. We agree with the reviewer that the selected carbon nanomaterials contain different functional groups on the surface that strongly affect the interaction between macrocycle. We believe that such a comparison of electroactivity on various nanomaterials is interesting and enhances the novelty of the present manuscript. We can speculate We speculate that the aromatic system of phthalimide offers additional π-π interactions between the porphyrazine and MWCNT attaching the Pz more firmly on the surface of the modified electron and thus efficient electron transfer. However, this interesting point have to be studied more by using other physicochemical techniques in future works. An additional comment has been added to the revised manuscript.
2. Line 59, the author claim that “Although the electrochemical properties of nickel (II) containing porphyrazines (Pzs), especially sulfanyl porphyrazines, are promising, there (they?) are scarcely studied”. Can the author clarify more why sulfanyl are promising or why they are less studied? Any specific role of N- ethylphthalimide in Pz3?
Porphyrins and phthalocyanines are the macrocyclic compounds that have been studied in much more detail, while the porphyrazins are a group that has received less attention so far. The presence of eight thioether moieties attached directly to the macrocyclic ring changes the properties of the macrocycle. As compared to other porphyrazine derivatives, sulfanyl Pzs offer high yield of synthesis, are very stable chemically (and are photostable as well). The reason for the fact that they are less studied is the high cost of the precursor and ironically their relatively easy preparation. Sulfanyl porphyrazines apart from rare exceptions are prepared from dimercaptomaleonitrile derivatives that can be obtained by alkylation of dimercaptomaleonitrile. Because of that, the spectrum of possible compounds is strongly restrained. Atop of that, symmetrical sulfanyl porphyrazines often offer very little or no interesting photochemical features.
The substituent was chosen based on the phthalimide properties. When this structural motif directly fused to the macroring, is serves as the electron acceptor, improving the electrical properties, while its presence in the periphery of the porphyrinoid results in photo-response behavior with potential applications in molecular electronics.
3. Another concern for Raman spectrum in Figure S1 is the superlattice, charge-transfer may cannot well interpret the drastic Raman shift of E2g mode in superlattice, such a phenomenon is drastically different from the reported Raman spectra in layered superlattice, (J. Am. Chem. Soc. 2017, 139, 16398−16404; Adv. Mater. 2020, 1907645; Nature 555, 231–236 (2018).) the authors shall check each Raman peak carefully and explain more.
4. Line 64 shall be their advantages are …
It was corrected
5. Line 163, please provide more details of Q band.
We extended the part of manuscript regarding Q band.
6. The caption in Figure 4 is misleading, the author should clarify the specific measurement condition for each CV spectrum. Moreover, the insets in Figure 4D, 4E, 4F are too small.
It was corrected

Reviewer 2 Report
The authors present the preparation of a new nickel porphyrin complex that is characterized and then used to modify several supports. The electrochemical behavior of these modified materials is analyzed, and the electrochemical reactivity of these materials with hydrogen peroxide is explored. I found the study to be well done overall with one minor caveat, the text clear, and the results supported by the data presented. This is a timely contribution to the field. I have only two suggestions.
1) I don’t have a sense from the manuscript or SI how much nickel was adsorbed on to the surface of the materials. In principle, the electrochemistry can tell this and while the data presented suggest this would be accurate, validation of that would be useful. Thus, I recommend a UV-vis of the nickel solution after dipping to indicate maximum concentration of adsorbed nickel complex.
2) For the initial CVs of modified materials, it would be useful to have the CV of the nickel compound for comparison. Even referring to the voltammogram in the Supporting Information would help a reader.
Author Response
The authors present the preparation of a new nickel porphyrin complex that is characterized and then used to modify several supports. The electrochemical behavior of these modified materials is analyzed, and the electrochemical reactivity of these materials with hydrogen peroxide is explored. I found the study to be well done overall with one minor caveat, the text clear, and the results supported by the data presented. This is a timely contribution to the field. I have only two suggestions.
1) I don’t have a sense from the manuscript or SI how much nickel was adsorbed on to the surface of the materials. In principle, the electrochemistry can tell this and while the data presented suggest this would be accurate, validation of that would be useful. Thus, I recommend a UV-vis of the nickel solution after dipping to indicate maximum concentration of adsorbed nickel complex.
Thank you for this valuable comment. The amount of electroactive compound on the surface is very important. We were able to estimate the nickel loading by using CV method. We calculated the charge under the Ni2+/Ni3+ redox peak and according to the data we estimated the mass of nickel that is equal to 11.1 ng. Unfortunately, we were not able to apply UV method, however, we believe that CV method is good for such as approximation.
2) For the initial CVs of modified materials, it would be useful to have the CV of the nickel compound for comparison. Even referring to the voltammogram in the Supporting Information would help a reader.
In the manuscript, we presented the CV response of GC electrode with an adsorbed thin layer of Pz3. The CVs were recorded in aqueous electrolytes. Please see Fig. 2. Such experiments allow us to verify the electrochemical activity of Pz3 in aqueous electrolytes